# Position: Quantum Deep Learning Still Needs a Quantum Leap

**Hans Gundlach** [1]   **Hrvoje Kukina** [2]   **Jayson Lynch** [1]   **Neil Thompson** [1]

## Abstract

Quantum computing technology is advancing rapidly. Yet, this position paper argues that even accounting for these trends, a quantum leap would be needed for quantum computers to meaningfully impact deep learning over the coming decade or two. We arrive at this conclusion based on a first-of-its-kind survey of quantum algorithms and how they match potential deep learning applications. This survey reveals three important areas where quantum computing could potentially accelerate deep learning, each of which faces a challenging roadblock to realizing its potential. First, quantum algorithms for matrix multiplication and other algorithms central to deep learning offer small theoretical improvements in the number of operations needed, but this advantage is overwhelmed on practical problem sizes by how slowly quantum computers do each operation. Second, some promising quantum algorithms depend on practical Quantum Random Access Memory (QRAM), which is underdeveloped. Finally, there are quantum algorithms that offer large theoretical advantages, but which are only applicable to special cases, limiting their practical benefits. In each of these areas, we support our arguments using quantitative forecasts of quantum advantage for current quantum algorithms that build on the work by Choi et al. (2023) as well as new research on limitations and quantum hardware trends. Our analysis outlines the current scope of quantum deep learning and points to research directions that could lead to greater practical advances in the field.

## 1. Introduction

Computing hardware has played a crucial role in the development of deep learning. The utilization of GPUs for machine learning tasks has been cited as the catalyst point for the deep learning revolution (Garisto, 2024). Indeed, the famous "bitter lesson" is that the most effective methods in artificial intelligence are those that best leverage large amounts of compute (Sutton, 2019). This trend has driven enormous increases in computing investment in artificial intelligence, which is beginning to hit the limits of our computational capacity (Sevilla et al., 2024). Quantum hardware has theoretical advantages in domains like cryptography and chemistry (Dalzell et al., 2023). Naturally, one might wonder if quantum hardware could lead to a paradigm shift in artificial intelligence similar to the increase in classical processing power in the past. There is a wide range of quantum algorithms for machine learning tasks (Biamonte et al., 2017), but how useful they are is often difficult to evaluate (Bowles et al., 2024). There are significant challenges to evaluating quantum's potential for accelerating deep learning. First, many quantum algorithms are not directly analogous to their classical computing counterparts. For instance, they often do not provide as much information as their classical counterparts or they can depend on incredibly specialized conditions that are hard to assure in practice. Second, quantum hardware is so much slower than classical hardware that theoretical runtime advantages can be lost (Choi et al., 2023). Third, many quantum machine learning algorithms depend on QRAM which faces large technical difficulties (Jaques & Rattew, 2023). Trends in quantum hardware will alleviate some of these difficulties but many will remain unaddressed without new breakthroughs. Hence, we argue that **quantum deep learning needs a quantum leap**. In the process of building our argument, we identify applications of quantum computing in the deep learning pipeline and outline their challenges. In addition, we extend the forecasting model from Choi et al. (2023) to understand the implications of quantum hardware trends. Our research on quantum computing trends used as inputs into this model are presented in the Appendix (see Appendix C). We hope this investigation helps realistically illustrate the quantum machine learning landscape, its potential applications to deep learning, and the challenges it faces. We also hope our investigation helps researchers in quantum computing and machine learning better focus on research likely to yield practical advantages. Table 1 (after our advantage-model setup) previews our findings across the deep learning pipeline; each row links to the section where we develop the argument in detail.

---

[1]MIT FutureTech, CSAIL Cambridge, MA, USA [2]TU Wien, Vienna, Austria. Correspondence to: Hans Gundlach <hansgund@mit.edu>, Neil Thompson <neil_t@mit.edu>.

*Proceedings of the $43^{rd}$ International Conference on Machine Learning*, Seoul, South Korea. PMLR 306, 2026. Copyright 2026 by the author(s).

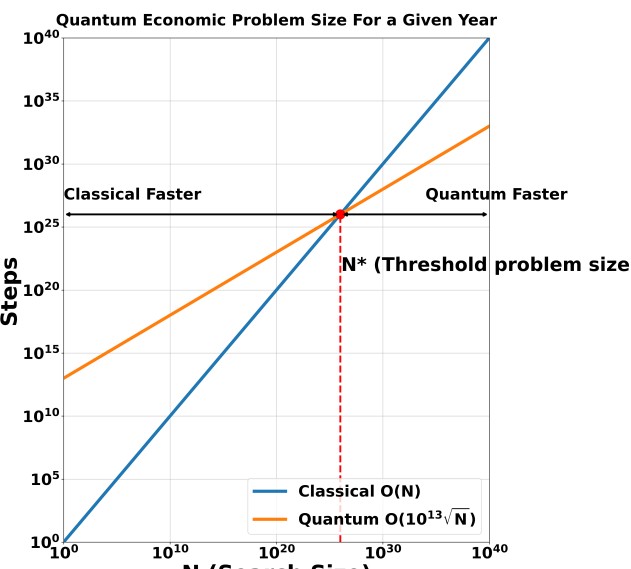

*Figure 1.* Quantum Economic Advantage (QEA) Problem Size in a given year. Quantum computers come with large constant hardware overheads but algorithmic advantages. Therefore, problem sizes must be over a critical threshold to have an advantage on a quantum computer.

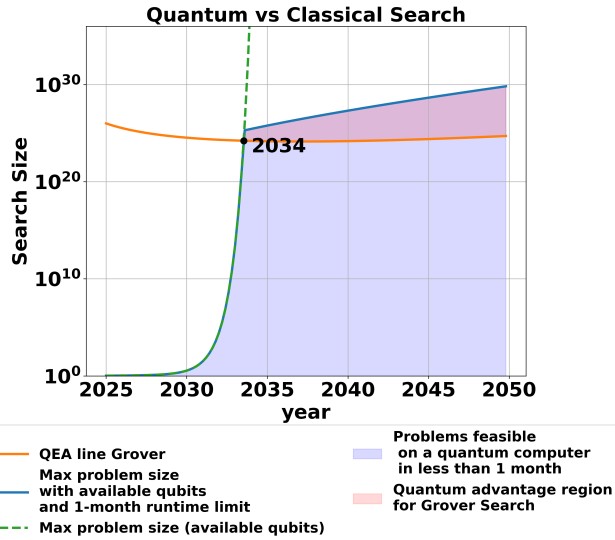

*Figure 2.* QEA threshold problem size over time taking into account hardware trends in quantum-classical overhead. The green line represents the maximum theoretical size that could be achieved without any time limit. We also choose to include a time limit of 1-month to better model constraints in machine learning. The quantum advantage region illustrates problem sizes that are feasible and preferable to run on a quantum computer in comparison to a given classical algorithm.

## 2. Quantum Advantage Model

Our forecasting model is based on the quantum economic advantage framework from Choi et al. (2023). This framework accounts for the fact that not all problem sizes are feasible to do on a quantum computer. First, if quantum computers have modest asymptotic advantages but large hardware overhead, large problem sizes are necessary to show speedups compared to classical computers. For instance, given an current overhead of $10^{13}$ factor slowdown [1], a quantum algorithm for a problem of size $N$, with a runtime of $\sqrt{N}$ quantum operations must be used on problem size greater than $10^{26}$ to have speedup over a classical algorithm with a runtime of $N$ classical operations ($10^{13}\sqrt{x} = x \Rightarrow x = 10^{26}$). This is represented by the intersection point (the Quantum Economic Advantage Point (QEA) or simply the advantage point) in Fig 1. Second, there must be a significant number of qubits to actually execute the algorithms. This is represented by the qubit feasibility curve in Fig 2. Third we introduce a time constraint to account for the fact that machine learning algorithms (particularly) subroutines should not take more than a month (complete training runs should not take more than about a year (Sevilla et al., 2022)). The

number of available qubits as well as the hardware overhead evolves with time. We account for this by completing an investigation into quantum computing trends and developing a model of how these trends affect logical qubit numbers and slowdowns. These trends and limitations outline the space of feasible problems for a given algorithm on quantum computer over time shown in Fig 2. A full description of our model as well as our analysis of quantum hardware and trends is in Appendix B.

## 3. Quantum Computing for Data Preprocessing

### 3.1. Data Selection and Processing for Deep Learning

Deep learning is fundamentally built on learning from large quantities of data. Large language models in particular, are on the verge of using almost all data that is on the web (Villalobos et al., 2024). Future models will not only be based on large web corpora but on large quantities of synthetic or experiential data (Ord, 2025). These large datasets necessitate data pruning and processing to be efficiently used to train deep neural networks. Modern state-of-the-art models use clustering of massive datasets to identify duplication and select relevant data (Grattafiori et al., 2024). Proper selection can also lead to large gains in training performance. A particularly good example is Sorscher et al. (2022), which uses k-means clustering on the embedding space of Ima-

---

[1]For some superconducting devices we have a $\sim 10^3$ slowdown in gate-speed vs floating point operation, $\sim 10^2$ factor overhead from error correction(Choi et al., 2023), and a $\sim 10^8$ factor more GPU parallelism per dollar (Helsel, 2022; Amazon Web Services, 2025). See Appendix C.1

| DL stage | Best quantum option | Speedup | Status |
|---|---|---|---|
| Data processing (§ 3) | q-means; quantum PCA | Exponential | Blocked by QRAM |
| Hyperparameter and Action search (§§ 4.1–4.2) | Grover; Dürr–Høyer | Quadratic | Infeasible ($N > 10^{24}$) |
| General Matmul / Attention (§ 5) | Swap-test; Gao attention | Polynomial | Lost to overhead, QRAM restrictions |
| NN training (classical) (§ 5.4) | Allcock; Liu; Newton; Zlokapa | Mixed | Promising variety, but slow / restrictive constraints |
| Quantum NNs / VQAs (§ 6) | VQAs; kernel methods | Unclear | No advantage in benchmarks |

*Table 1.* Status of quantum acceleration across deep learning pipeline stages. Each stage links to the section detailing the argument.

geNet. With this clustering, they were able to selectively prune datasets and train models much more efficiently. This data pruning allows them to train with exponential rather than power-law scaling with pruned dataset size. Data selection for state-of-the-art models often includes clustering of massive amounts of data (Grattafiori et al., 2024).

### 3.2. Quantum Clustering

There are many quantum clustering methods. These include quantum supervised clustering (Lloyd et al., 2013), quantum k-medians clustering (Getachew, 2020), and q-means (Kerenidis et al., 2019). However, many of these methods are not exactly analogous to their classical clustering. Q-means is a quantum variant of k-means that is able to both cluster data points as well as return the centroids of the resulting clusters in $\widetilde{O}(k^2d + k^{2.5})$ time compared to classical $O(ndk)$ where $k$ is the number of clusters, $n$ is the number of samples in training set, and $d$ is the dimension of features used in clustering (Kerenidis et al., 2019). This method has exponential advantage in terms of the dataset size $n$ but relies on the notion that the dataset is "well-clusterable" and requires QRAM. If these two conditions are met then clustering looks like a promising method on a quantum computer. Fig 3 shows the date when such an algorithm would have an advantage over a classical computer in addition to the sizes necessary to find such an advantage. In general, we find constants not important for algorithms with exponential advantage so we model the comparison as $O(n)$ vs $O(\log(n))$. This is an enormous advantage if tasks involve datasets with a large number of points. For instance data analysis on large language model datasets which can contain $10^{13}$ tokens currently and are projected to grow more than 2x per year at least for the next few years (Villalobos et al., 2022).

### 3.3. Other Methods: PCA, Perceptron, SVM

In general, quantum methods are particularly good for data analysis methods where only limited output is desired and which depend on linear algebra subroutines like matrix inversion or factorization. This is due to the nature of the quantum algorithms, in particular the Harrow-Hassidim-Lloyd (HHL) algorithm, which is used as a subroutine to many data analysis tasks (see Appendix F). For HHL, measuring the resulting answer state is hard and getting $N$ components often involves $O(N)$ overhead. Particular problems that solve this constraint are linear regression, where the computational bottleneck is on matrix inversion and where the output is a select set of coefficients. In addition, quantum methods exist for PCA (Lloyd et al., 2014), linear regression (Wang, 2017), SVM (Rebentrost et al., 2014), Online Perceptron (Wiebe et al., 2016), and topological data analysis (Schmidhuber & Lloyd, 2023). We are not familiar with the current state of the art models that employ these methods in processing. Yet, methods like PCA used to be an important step in processing images for supervised learning algorithms (Ng & Le, 2013). If these data analysis and preprocessing methods became significantly cheaper to run (whether classically or from a quantum speedup) it seems likely they will find uses in the deep-learning pipeline.

## 4. Grover's Based Optimizations: RL and Hyperparameter Search

### 4.1. Quantum Hyper-parameter Optimization

Grover's algorithm is a quantum algorithm that can find an element in an unstructured list in $O(\sqrt{N})$ time. It is also possible to find the maximum and minimum of an unstructured list in $O(\sqrt{N})$ time using the Dürr and Høyer algorithm (Durr & Hoyer, 1996). Can we use this ability to optimize deep neural networks? Tentatively, we think it is possible under certain conditions. First, hyperparameter search for $N^2$ initialization must be better than hyperparameter search using a different technique using $N$ initialization (i.e, SGD). Second, the loss of the network for a specific initialization must not be stochastic since Grover's or similar algorithms easily fail with a stochastic oracle (Regev

& Schiff, 2008). Third, if we do not have QRAM it must be possible to prepare a coreset or sample of data that can be used as a proxy for loss on the entire dataset (Harrow, 2020). Translating a large classical dataset into a quantum computer will lose much of the quantum advantage. However, the experiment sizes necessary to see advantage with Grover's algorithm look implausibly large (see Fig 3b). For instance, the largest hyper-parameter search we know of in deep learning did several hundred experiments (Britz et al., 2017), whereas we find that a search size of $10^{24}$ is necessary to see an advantage at the QEA year of 2034. There still might be hope to use such methods to search for optimal weights in binary neural networks (Jura & Udrescu, 2025). The search space here could be potentially very large. Deepseek V3 has $6.85 \cdot 10^{11}$ parameters (DeepSeek-AI, 2024) and models are getting larger.

## 4.2. Quantum Action Selection for Reinforcement Learning

Reinforcement learning involves agents who interact and make choices in a given environment. Many RL algorithms involve searching through a range of choices to find optimal actions. For instance, Q learning, involves choosing an action a that maximizes the Quality function for a given state. These sorts of selection problems can be quadratically faster on a quantum computer(Dunjko et al., 2016) (Dong et al., 2008)(Saggio et al., 2021). Similarly, results have shown that quantum agents could have a quadratic speedup in some multi-arm bandit scenarios (Wang et al., 2021). However, as is the case with Grover-based speedups, very large problem sizes ($> 10^{24}$) are necessary to see advantage, which makes such optimization impractical for foreseeable RL problems (see Fig 3(b)).

# 5. Quantum Linear Algebra: Matrix Multiplication and Enhanced Training

## 5.1. Quantum Matrix Multiplication

We compare against classical algorithms as actually used in deep-learning practice — e.g., standard $O(N^3)$ matrix multiplication rather than Strassen's $O(N^{2.8})$, and standard quadratic attention rather than linear variants — since these faster theoretical alternatives are rarely deployed in production.

Matrix multiplication is a central computational component of most modern deep-learning architectures across training and inference. Classical matrix multiplication is a well-studied and highly optimized problem. There exist three conventional classical algorithms for dense square n by n matrix multiplication. These include the naive algorithms that run by multiplying rows by column vectors to get every entry in the result matrix, running in $O(N^3)$. For much

larger matrices, we can employ Strassen's method, which run in $O(N^{2.8})$ (Hartnett, 2021) but comes with significant overhead. There are also laser-based methods in combination with innovations from Coppersmith-Winograd that run in $O(N^{2.37})$ (Nadis, 2024). Yet, many approaches beyond Strassen are never used in practice and are only of theoretical interest (Le Gall, 2012). Most researchers in the field believe that matrix multiplication could eventually run in close to $O(N^2)$ time (Robinson, 2005). However, no such classical algorithms exist at present. Quantum algorithms open up the possibility to reach and even exceed this bound. Nevertheless, many quantum matrix algorithms come with key limitations that limit their practical usability. For instance, all the methods we consider use QRAM if the matrix entries are given as classical data (see Appendix A).

For general dense N by N square matrices, "swap-test" quantum methods can multiply matrices within epsilon accuracy in time $\widetilde{O}(N^2/\epsilon)$ (Bernasconi et al., 2024). Furthermore, for matrix multiplication of the form $AB$ where $A$ is $N$ by $M$ and $B$ is $M$ by $N$ with $w$ nonzero elements in the output matrix and $w < \sqrt{N}$ matrix multiplication can be done in $O(M \log(N)N^{2/3}w^{2/3})$ time (Buhrman & Spalek, 2004). This becomes $\widetilde{O}(N^{5/3})$ for square matrix multiplication with poly-logarithmic nonzero output entries. If we want to multiply two dense square matrices but only care about one aspect of the output i.e, one entry of the output or the expectation value of some quantity then in this case dense "multiplication" can be done in time $O(\sqrt{N}\log(N)k^2/\epsilon)$ (Wossnig et al., 2018), where $k$ is the condition number. Direct HHL (Harrow-Hassidim-Lloyd) can do a summarized form of matrix-vector multiplication in $O(\text{polylog}(N))$ and hence return statistics on matrix matrix multiplication in $\widetilde{O}(N)$. However, this method includes many assumptions that are unlikely to be met by current or future deep learning problems (see Appendix F).

We focus our analysis on dense square matrix multiplication with full matrix output, which on a quantum computer runs in $\widetilde{O}(N^2/\epsilon)$ (Bernasconi et al., 2024) time, and compare to classical $O(N^3)$ approaches. This is the most common case for classical deep learning frameworks. Modern AI models also leverage structured sparsity to improve matrix multiplication outcomes, but we know of no quantum algorithms that could yield advantageous outcomes in these cases with the same output. Given the theoretical asymptotic benefits of quantum matrix multiplication, could these methods be implemented in practice? We find that the overhead due to quantum computing makes it infeasible to use this method for practical matrix multiplication. This holds even taking into account future progress in superconducting quantum computing. Our conclusion could change if other fast approaches like photonic quantum computing are developed along with the corresponding development of QRAM.

In regard to the greater "zoo" of quantum matrix multiplication algorithms, these look unapproachable as subroutines in classical neural network matrix multiplication. Yet, they could form important subroutines in fully quantum neural networks. These networks face other issues related to implementing activation functions along with these subroutines (Schuld & Petruccione, 2022). Yet, this remains an intriguing area for further research.

## 5.2. Matrix Vector Multiplication

Curiously, since matrix-vector multiplication only requires outputting a vector instead of a matrix as in matrix-matrix multiplication, such an operation has a larger advantage with quantum computers. Classical computers can do this operation in $O(N^2)$ time. While using HHL (along with the corresponding conditions) could be done in $O(N)$ time. However, the advantage of such a method in conventional deep-learning workflows looks limited because with multiple vectors, i.e., batch processing, classical computers can leverage more efficient matrix operations. Nevertheless, efficient matrix-vector multiplication is useful for neural network training methods (see Section 5.4).

## 5.3. Quantum Attention Mechanism

One of the largest current bottlenecks to transformers is the large size of attention matrices needed. Given a sequence of $N$ tokens, attention theoretically requires creating and multiplying a matrix of size $N^2$. If we assume that the attention matrix has at most $k$ nonzero entries per row, and the query/key dimension is $d$ then the Attention mechanism can be computed in time $\widetilde{O}(N^{1.5}k^{0.5}d + Nkd)$ compared to classical $O(N^2d)$ (Gao et al., 2023). Given the limited asymptotic benefit, we find the overhead makes it impossible to find any practical benefits. This has the same failure point as matrix multiplication, where viable sizes are impossible to do in a reasonable time (see Fig 3(c)).

## 5.4. Quantum Computing for Classical Neural Network Training

There are several proposals to train neural networks using quantum computers. Many of these proposals are based on using quantum linear algebra primitives like HHL to simulate or solve different aspects of the training procedure in neural networks. Unfortunately, most of these proposals suffer from a list of caveats (like HHL see Appendix F) that most likely prohibit practical application. Liu et al. (2024) proposes a method to train a neural network in time $O(T\text{poly}(\log n, \frac{1}{\epsilon}))$ of parameter size n, number of iterations T, and precision $\epsilon$. However, this proposal relies on a complicated set of conditions, which include constant sparsity of weights, "dissipation" and small learning rates. Another proposal based on using quantum inner product

calculations can train and do inference in asymptotically better time if the network has many edges. Specifically, this approach considers a network using an inner product which is $\epsilon$-approximate with probability $1 - \gamma$ where $M$ is the number of input samples, $N$ is the neuron number, $E$ is the total number of edges in a given neural network, and $R$, $R_e$ are factors that depend on network and data and are small for practical purposes. This network can be trained in time $\widetilde{O}\left((TM)^{1.5}N\frac{\log(1/\gamma)}{\epsilon}R\right)$ and inference can be done in time $\widetilde{O}\left(N\frac{\log(1/\gamma)}{\epsilon}R_e\right)$ This is in comparison to $O(TME)$ and $O(E)$ respectively for classical neural networks (Allcock et al., 2019). Current neural networks, like llama 3 405B have a model dimension of 16834, which in a dense network means each neuron has around 16834 edges for each neuron (Grattafiori et al., 2024). However, even with increasing neural network size and increasing quantum speed, this is likely not enough to overcome the current quantum slowdown of $10^{13}$ (see Appendix C.1).

Another plausible use case is the application of Newton's method to neural network training. Newton's method is impractical in current use cases because it requires inverting a Hessian matrix of size $N^2$ where $N$ is on the order of the number of parameters in the network. Speculatively, this inversion could be done much faster using quantum linear algebra techniques, possibly exponentially faster (Rebentrost et al., 2018). In practice, considering the time necessary for reading in and out of classical data, the iteration complexity is $\widetilde{\Omega}(N)$. This may yield a speedup relative to classical Hessian methods; however, much more work needs to be done to show an advantage over classical gradient descent (Zhang & Shao, 2024).

One tantalizing proposal is based on the approximation of wide and deep neural networks. In this regime, neural networks are well approximated by kernel methods (Zlokapa et al., 2021). This method relies on using HHL to invert the neural tangent kernel matrix K between any two data points in the training set. Such a method has the possibility of $O(\log(n))$ training where $n$ is the number of data points in the training set. However such a method relies on HHL and QRAM along with its corresponding conditions (see Appendix F), which puts restrictions on the dataset. Zlokapa et al. (2021) verify that these restrictions are met with MNIST. However, we are unsure if this can be applied to general machine-learning problems. Furthermore, to the best of our knowledge, such a method does not return trained weights for a neural network that can be used classically; it simply provides the answer of a trained neural network and therefore cannot be formally called a "training" method.

We are highly skeptical of methods proposing exponential training advantage given the complicated conditions. However, if these methods' conditions are satisfied along with

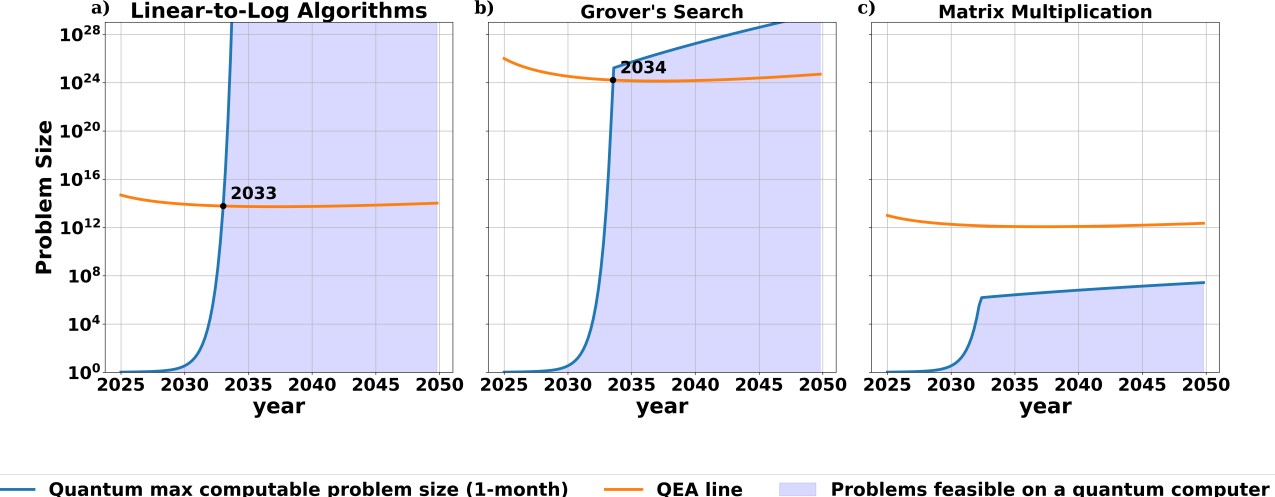

*Figure 3.* Three scenarios for quantum machine learning. Problems below the blue feasible line and above the orange economic advantage (QEA) line are likely to be impacted by quantum computing. Fig 3a illustrates the case where a quantum algorithm has an exponential speedup over a linear classical algorithm. Fig 3b captures the problem size necessary to see advantage with Grover's speedup. Fig 3c shows the quantum advantage diagram for matrix multiplication, which demonstrates that while potentially feasible (with QRAM), it will not be advantageous on a quantum computer for the near future.

QRAM, then they could be promising for deep learning training (see Fig 3a).

# 6. Quantum Neural Networks

So far, we have addressed quantum computers as machines capable of implementing subroutines for classical systems. However, given the expressivity of quantum computers to solve problems, could we instead try to optimize quantum-based neural networks rather than their classical counterparts? Quantum neural networks try to train and improve a network of quantum layers using a quantum or classical computer. Like classical neural networks, they have a series of layers (in this case unitary blocks) which are parametrized by some trainable parameter $\theta$ (Cerezo et al., 2021). Many current quantum neural networks are an instance of Variational Quantum Algorithms (VQAs).

## 6.1. Variational Quantum Algorithms

Variational quantum algorithms are a highly researched class of quantum machine learning methods. Notably, these methods can be run on NISQ (noisy intermediate-scale quantum computers), which makes them a promising quantum benchmarking target. These methods use a quantum computer to generate states, while measurement and optimization are done on a classical computer. However, such methods can be especially hard to characterize and do not generally have asymptotic times. Further, comprehensive recent work has found little or no advantages for these methods when prop-

erly benchmarked (Bowles et al., 2024).

## 6.2. The Potential of Quantum Neural Networks in General

Quantum neural network training, depending on architecture, can be quite difficult and suffer from poor gradient information due to Barren Plateaus (McClean et al., 2018). Unfortunately, many current neural network architectures have no clearly specified formal bounds on training. Therefore, much of our analysis based on asymptotic run-times does not apply. It is possible to simulate some of these architectures. Nevertheless, recent benchmarking studies have shown no advantage for many quantum neural networks (Bowles et al., 2024). However, much of modern machine learning is an experimental field. Quantum machine learning is at an inherent disadvantage as simulations are very limited without full-scale quantum computers. Will quantum neural networks catch up? At this moment, it's hard to say. We give some intuition for and against quantum neural networks.

WHY WOULD QUANTUM NEURAL NETWORKS BE BETTER?

1. N qubits can generally encode $2^N$ pieces of data (Giovannetti et al., 2008). This gives them a memory advantage. Further, this may lead to reduced data and parameter transfers, leading to better runtime for distributed algorithms (Gilboa et al., 2024).

2. Quadratic and sometimes exponential speedup for specialized matrix operations using algorithms like HHL, and density matrix exponentiation (Schuld & Petruccione, 2022)(Biamonte et al., 2017).

3. Possible quadratic improvements in sample efficiency and parameter estimation (Montanaro, 2015).

WHY WOULD QUANTUM NEURAL NETWORKS BE WORSE?

1. Quantum computers generally have many restrictions, like no-cloning, and state-collapse under measurement, which may limit data transfer, data use, and data reuse.

2. For classical data, classical to quantum data transfers can be large enough to overwhelm much of the quantum advantage (Aaronson, 2015)(Schuld & Petruccione, 2022).

### 6.2.1. QUANTUM LEARNING FROM QUANTUM DATA?

A large issue in quantum machine learning is the problem of reading in and outputting large amounts of classical data. This is the issue that plagues the usage of quantum subroutines in classical network processing. Therefore, it's natural to consider deep quantum networks that interact minimally with classical computers during their processing. Huang et al. (2022) found that quantum computers have an exponential advantage in learning from quantum experiments. However, it is unclear these advantages will carry over into classical problems like language generation or image detection, so their scope may be extremely limited to topics like physics and chemistry research.

## 7. Other Approaches in Quantum Machine Learning

Our paper concentrates on methods that are fault-tolerant, have at least approximate asymptotic times, and have viable use in the current or future deep learning pipeline. However, this is a rather limited perspective on the variety of quantum machine learning methods available. Here we have a brief outline of these other methods.

### 7.1. Quantum Kernel Methods

These methods rely on mapping data into a higher-dimensional hilbert space using a quantum circuit. This can serve as a feature map which is fed into another quantum circuit to compute the inner product. This product is then fed into a classical kernel machine (i.e., SVM)(Schuld & Petruccione, 2022). QSVM is a prime example of such methods (Rebentrost et al., 2014). These are promising techniques. However, we do not know any uses of such techniques in the deep learning pipeline.

### 7.2. Other Methods

An important limitation of our paper is that we can only examine methods where sufficient information is known. The set of all methods that have been suggested is much larger. Of the ones we have not included here, some are relatively more developed/discussed (e.g., Quantum Annealing, Quantum Reservoir Computing (Kobayashi et al., 2024), Quantum Generative Modeling (Nath et al., 2021)) and others much less so.

## 8. Robustness

In this section, we ask what changes would affect the year we see quantum advantage for the problem in our study? These studies also offer a perspective on the sensitivity of parameters in our model. Here we vary the rates of growth in our model. We also vary the classical and quantum constants by multiplying the runtime complexity with factors from $10^{-3}$ to $10^4$. The results of our analysis are in Fig 4. In general, we see that the advantage year does not depend significantly on many growth rates (physical qubit, decrease in quantum gate time, etc). However, a decrease in quantum gate time, or equivalently, the rate of quantum speed improvement (neglecting error correction), does have a significant effect on the possibility of quantum matrix multiplication. Yet, beyond Moore's law, improvements in quantum gate speeds are necessary for this to be realized. Our analysis for Grover's algorithm is sensitive to our estimates of overhead/algorithmic constants. This sensitivity is one-sided. Estimates that are more favorable to classical computing, like increasing quantum constants, change the advantage year significantly, while using parameters that are more favorable to quantum computing has little effect on the advantage year for Grover. Quantum algorithms that are sensitive to this effect are bottlenecked by the maximum quantum running time. The predicted rate of gate speed increase is significantly slower than the increase in problem size available to solve on a given number of qubits (see Fig 2). Since many of the algorithms we address are time-bound, not qubit-bound, the growth rate of physical qubits has little effect. In addition, we see that reasonable variation in the maximum time limit allowed does not change our predictions (this is shown in Appendix D).

## 9. Alternative Views

First, we would like to reiterate the limitations of our model used to make our argument. We are extrapolating from current trends in superconducting qubits (we analyze other types of quantum computers in Appendix E) . We assume algorithmic constants equal to 1 (However, our model is robust to large variations see Section 8). We focus on algorithms with clear asymptotic times and assume these asymptotic

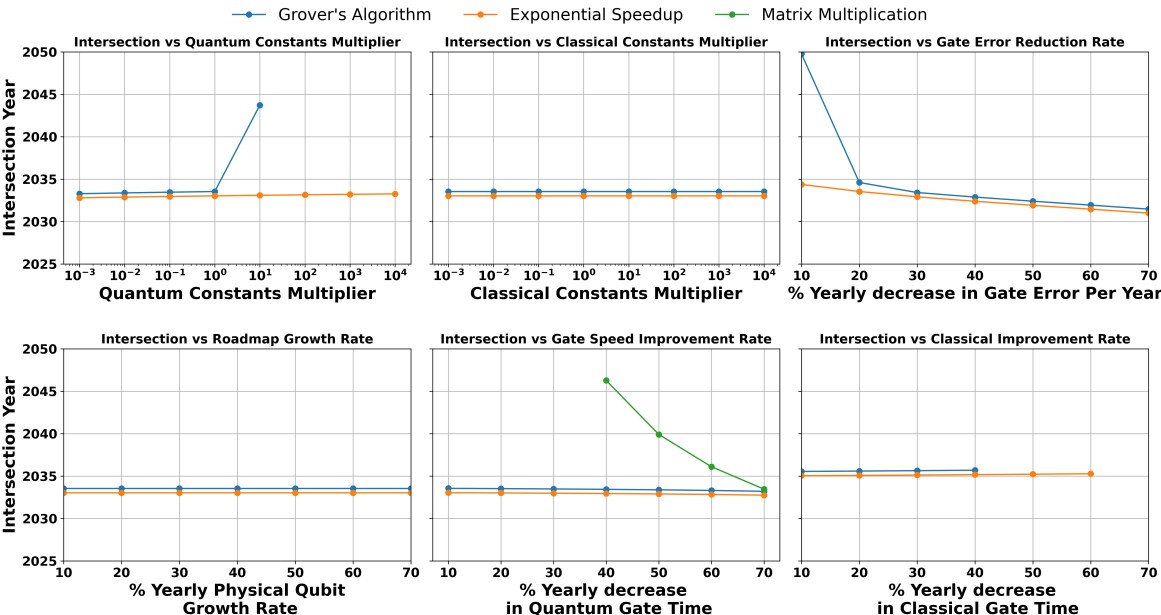

*Figure 4.* Graphs of how the first year of Quantum Economic Advantage changes as model parameters are varied individually for each algorithm class measured. QEA years past 2050 are not plotted, which is almost always the case for dense matrix multiplication. In general, our QEA predictions do not change substantially with significant changes in algorithmic constants or rate trends.

times do not change. Variations in many of these assumptions lead to plausible alternative views.

**If You Build It, They Will Come?** At the moment, the potential future applications of quantum computers are relatively specialized and limited (Castelvecchi, 2024). Yet, the development of fault-tolerant quantum computers may radically change the speed and landscape of quantum algorithms available. New hardware could unlock the ability to rapidly test and develop new quantum algorithms, leading to faster progress.

**Continued Trends?** Classical computing performance in the form of GPUs has increased significantly (Hobbhahn & Besiroglu, 2022b). However, trends in GPU performance may also eventually stall in the 2030s (Hobbhahn & Besiroglu, 2022a). This would allow quantum computers to gain a larger advantage over classical computers. Nonetheless, other classical technologies beyond GPUs like optical computing or neuromorphic computing, may take their place.

**Alternative Algorithms and NISQ Computing** Our analysis deliberately focuses on fault-tolerant quantum algorithms with established asymptotic complexity, where rigorous claims can be made across broad problem classes. This scope has two implications. First, no medium- or large-scale fault-tolerant quantum computer currently exists, so

direct empirical validation at problem sizes of interest is infeasible—we rely on asymptotic analysis instead. Second, our model cannot address algorithms without clear asymptotic times, such as Variational algorithms (see Section 6.1) and other NISQ approaches, which may eventually show advantage even if the methods we focus on do not. Intermediate-scale quantum computers with noisy qubits will be available much sooner than fault-tolerant ones, and there is substantial work on useful computation without the benefits of error correction (Bharti et al., 2022). Existing empirical benchmarking of these near-term methods has largely found negative or mixed results (see Section 6.1), and to our knowledge none are currently deployed in industry.

## 10. Conclusion and Call to Action

We've outlined different approaches to applying quantum computing in deep learning. Quantum computers are improving fast, yet these hardware trends will not be enough to see the practical utility of quantum benefits in deep learning. How can we address this shortfall? As we have outlined, many approaches to quantum deep learning have a common set of pitfalls. Achieving breakthroughs in these areas would bring quantum deep learning closer to a practical reality.

**Accelerate QRAM Development:** Many applications of quantum computing in deep learning depend on loading and reading classical data, which depends on QRAM. Particularly those for large scale data analysis and processing

methods. Developing fault-tolerant QRAM may be an engineering challenge on par with building a fault-tolerant quantum computer or infeasible (Jaques & Rattew, 2023). We encourage much more work into developing QRAM hardware as well as QRAM-aware quantum algorithms.

**Develop QRAM-Free and QRAM-Aware Algorithms:** Given the formidable engineering challenges facing practical QRAM, algorithms that either avoid QRAM entirely or remain useful under realistic QRAM constraints are particularly valuable. Recent work such as Zhao et al. (2026) is a promising example: it establishes exponential quantum advantage for certain classical data-processing tasks using streaming quantum oracle access rather than full-scale QRAM. Likewise, Harrow (2020) demonstrates a quadratic speedup for hyperparameter search by working with coresets, sidestepping the need for QRAM-based data loading. More broadly, we encourage authors to present both QRAM and non-QRAM variants of their algorithms, in the spirit of Liu et al. (2024), who report asymptotic runtimes for their neural-network training procedure under both assumptions. This practice makes it far easier for the community to assess which conclusions survive once QRAM is removed from the picture.

**Consider Classical Speedups Relative to Quantum Computers:** The large overhead associated with quantum computers often makes the use of quantum algorithms unattractive even for problems with polynomial speedup. These include quantum matrix multiplication, quantum attention mechanism, and quantum search. Given trends in quantum hardware, this gap looks unlikely to be bridged by superconducting systems unless classical computers stall considerably for parallel operations (as they did for clock speeds (Rupp, 2018)). Yet GPUs, which offer greater scientific and parallel computation ability, have been improving rapidly (Hobbhahn & Besiroglu, 2022b). While GPU progress may eventually plateau (Hobbhahn & Besiroglu, 2022a), emerging technologies such as optical computing or neuromorphic computing could serve as alternatives. Photonic quantum computers have the potential for faster quantum computation (Rudolph, 2023). However, these systems come with their own unique drawbacks (Gschwendtner et al., 2023). Therefore, it is important to focus on problems where quantum computers can have a meaningful advantage despite quantum overhead.

**We Need Better End-to-End Characterization:** There are some algorithms that may not depend on dramatically faster quantum computers. However, these come with many uncertainties and conditions of their own. These include proposals for exponential speedup in classical training, some variational quantum algorithms, quantum algorithms for training wide and deep neural networks, and other approaches. We have tried to investigate many of these algorithms. How-

ever, these either do not have well-defined bounds, or have bounds that depend on complex conditions which are hard to map to real-world problems. Further, besides variational algorithms, few of these algorithms or their claims have been discussed in subsequent literature. This is particularly pertinent given that many algorithms with supposed quantum exponential advantage have been shown not to have exponential advantage (dequantization) (Tang, 2019).

**Further Algorithmic Breakthroughs:** We saw the size of the asymptotic advantage for quantum algorithms play a significant role in the feasibility of speedups to deep learning from quantum computing. Future improvements to quantum algorithms could have dramatic impact on the landscape, though at the same time future advances in classical algorithms (e.g. as seen by the quantum-inspired classical algorithms (Tang, 2019)) may give us many of these predicted benefits without the difficulty of developing quantum hardware.

In general, exponential improvements reliably lead to quantum advantage. Surprisingly, it can often be better to polynomially improve algorithms from $O(N)$ or $O(\sqrt{N})$ rather than improve higher order classical algorithms like $O(N^6)$ algorithms to $O(N^5)$. This is because at the problem size necessary to see an advantage, running the quantum algorithm will take prohibitively long.

Further, the development of quantum algorithms often has positive spillovers for classical computation. Tensor networks, originally from quantum many-body physics, have been applied to classical neural network compression (Novikov et al., 2015), and Tang (2019) used quantum-inspired techniques to develop the first classical polylog-time recommendation algorithm.

This paper outlines a wide variety of potential applications of quantum computers to deep learning. In each case, we find substantial challenges that would need to be overcome for quantum computing to meaningfully impact deep learning in the coming decade or two. But, quantum computing is still in its infancy, and the development of fault-tolerant quantum computers could bring forward cascading breakthroughs that would allow these systems to provide real advantage. In that spirit, we focus on the concrete bottlenecks that most plausibly separate today's capabilities from that future. We hope that by identifying the bottlenecks to achieving these goals - and in many cases quantifying what would be necessary to overcome them - we can help make this leap.

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

## A. QRAM

Many quantum algorithms including quantum database search and quantum machine learning involve retrieving and manipulating data from memory. When we refer to QRAM we refer to a system that is able to load an item $m_j$ from memory in the following way: $\sum_j \alpha_j |j\rangle |0\rangle \xrightarrow{QRAM} \sum_j \alpha_j |j\rangle |m_j\rangle$ . However, implementing such a system on quantum computers efficiently poses a significant challenge. If memory retrieval takes on the order of $O(N)$ computational steps, where $N$ is the size of the database, then quantum algorithms lose much of their advantage. There are several proposed QRAM designs, but the most prominent is bucket-brigade QRAM (Arunachalam et al., 2015). This is a theoretical proposal that has logarithmic retrieval costs in the size of the database. Specifically, only $O(polylog(N))$ active gates (i.e, gates that require energy during memory retrieval) are needed. In addition, this system has an overall readout error of $p \cdot polylog(N)$ for a database of N items with a per gate error probability of p. However, this system still suffers from several key drawbacks. First, $O(N)$ passive gates are required, which is quite large for the scales necessary to see quantum advantage. Second, adding error correction to this system would negate the logarithmic advantage as error correction would require time complexity based on the total number of gates $O(N)$. Error correction is not necessary if the gate error rate is sufficiently low. However, for algorithms like Grover's search that require $O(\sqrt{N})$ memory queries, the error rate per gate would have to be $\frac{1}{polylog(N)\sqrt{N}}$ which would require gate errors many order of magnitude smaller than currently exist (current gate error rates range from $.1\% - 1\%$). However, algorithms

like quantum matrix inversion and many quantum machine learning algorithms require only $1/polylog(N)$ error which is feasible (Arunachalam et al., 2015). Finally, many suggested hardware implementations (Trapped Ions, Photonic Transistors, etc) of bucket-brigade QRAM still suffer from bad energy scaling $O(N)$ or other impractical hardware constraints which limit scaling (Jaques & Rattew, 2023). To be fair, classical memory requires at least $O(log(N))$ operations for retrieval and volatile memory like DRAM require energy scaling $O(N)$. Given that quantum advantage requires large problem sizes (Choi et al., 2023) and QRAM error requires small problem size, QRAM limitation might restrict some quantum algorithms to intermediate problem sizes around $2^{40}$ (Jaques & Rattew, 2023).

## B. Quantum Economic Advantage Model

Our model is built on the model introduced in Choi et al. (2023). Quantum computers can implement better algorithms that are not possible on classical computers. However, quantum computers come with a large slowdown. This means that it is only advantageous to use quantum computer at large problem sizes.

In order to find the problem size that is optimal for quantum computing we estimate an overhead due to parallelism, quantum error correction, and other slowdowns. For example, we can calculate the size necessary for quantum advantage in Grover's algorithm (quantum search algorithm) as follows. We use current hardware overhead, which we estimate to be around $10^{13}$ (see Appendix C.1).

$$10^{13}\sqrt{x} = x \Rightarrow x = 10^{26} \qquad (1)$$

However, the problem size necessary to see an advantage changes over time due to trends in the relative speed between quantum and classical computers. For instance, advances in gate fidelity, advances in quantum error correction could decrease the overhead. On the other hand, faster GPU progress or new classical processors could further increase the overhead. We omit algorithmic constants from both the classical and quantum algorithms for a number of reasons. First, we have little knowledge of algorithmic constants for quantum algorithms. Second, it is harder to optimize quantum algorithms without access to fault-tolerant quantum computers. We expect quantum algorithmic constants to decrease significantly with the rise of large-scale quantum computers. For example, recent advances in quantum chemistry algorithms have reduced overhead by a factor by many orders of magnitude (Günther et al., 2025). Third, the overhead due to quantum hardware factors are extremely large ( $10^{13}$). Algorithmic constants must be of a similarly large scale to alter the majority of our conclusions. Our model is based on superconducting qubit hardware. Superconducting qubits

are one of the most popular quantum computing paradigms (Ruane et al., 2025). There has also been significant work benchmarking and evaluating resource estimate for superconducting quantum computing (Sevilla & Riedel, 2020) (Babbush et al., 2021). Finally, superconducting qubits generally have faster two-qubit gate times than other systems like ion-trap, and neutral atom computers (Gschwendtner et al., 2023). Gate speed is a significant factor in our model for the feasibility of quantum algorithms with only polynomial speedups. Superconducting systems generally require more error correction due to poor gate fidelity, which adds overhead. However, we estimate superconducting qubits still have faster gate speeds than most quantum computing paradigms. Photonic could be faster (Rudolph, 2023), but generally have a range of complications and resource constraints that are less well known. See Appendix E for our analysis of other quantum hardware paradigms.

### B.1. Finding Feasible Problem Sizes

A problem size might be theoretically advantageous for a quantum computer, but not feasible with our currently available quantum computers. We determine feasibility using two metrics: the number of qubits available and the time available. We impose a 1-month calculation time limit for quantum problems. We estimate the time needed for our computation based on error correction overhead and gate time (see Appendix C.1).

How are quantum computers limited by qubits? For most problems in quantum machine learning, and all the problems we consider in this paper, the number of logical qubits needed for a problem of size n is $O(\log(n))$. In other words, a quantum computer with n-qubits can solve a problem of size $2^n$. We determine the number of logical qubits using trends in the number of physical (error-prone) qubits (see Appendix C). We then determine the ratio between logical qubits and physical qubits using trends in qubit error rates (see Appendix C.3).

## C. Trends in Number of Superconducting Qubits

One of the most important trends in our model is growth in the number of physical qubits (error-prone qubits). However, it can be difficult to identify a consistent trend for the number of qubits. Quantum computers face a quality-quantity tradeoff. Some providers choose to create more qubits with less fidelity and vice versa. We choose to compromise by taking the 90th percentile quantile regression trend in physical qubit numbers rather than trends in the largest machines. This has a similar growth rate to providers like IBM, which might be of a more consistent quality.

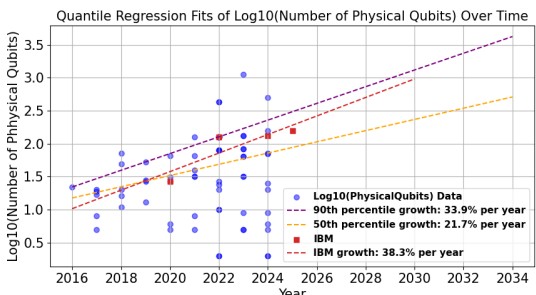

*Figure 5.* Graph of log number of physical qubits vs time for superconducting systems. Our default model is based on the 90th percentile quantile fit as this is reflective of mainstream providers like IBM. Data sourced from Ruane et al. (2025).

### C.1. Quantum Computers Have a $\approx 10^{13}$ Slowdown

Given the same financial and time resources, how much faster are classical computers on a per-operation basis? We assume superconducting hardware properties like processor speed using estimates from (Choi et al., 2023). Here we compare the number of gate operations executable on a quantum computer, given a budget of one dollar and 1 second of time vs the number of FLOPS possible on an H100 GPUs, given the same resources.

**Quantum Processor Ability** The cloud price for IBM's 27-qubit Falcon processor is $1.60\$$ per second (Helsel, 2022). Rigetti's superconducting processor has a similar price of $1.3\$$ per second (Microsoft, 2025b). Choi et al. (2023) gives superconducting clock speeds in the range of 2 MHz. The error correction overhead is assumed to give a slowdown of $10^2$. This leads to around $10^4$ logical operations per second with $1\$$ resources. However, on each qubit line we can implement gate operations so we can get roughly $10^5$ logical gate operations considering this form of parallelism.

**Classical Processor Ability** An NVIDIA H100 Tensor core can do around 2000 TFLOPS (FP16) (Corporation, 2024). Cloud prices for H100 GPUs are around 2-4 dollars per hour (Amazon Web Services, 2025). This means using a GPU we can get around $10^{18}$ operations per second with one dollar on a classical machine.

Comparing gate operations to Flops as is done in Choi et al. (2023) leads to an extreme quantum slowdown on the order of $\approx 10^{13}$.

### C.2. Trends in Quantum Computer Gate Time

Quantum computer gate times are usually much longer than classical transistor switching speeds. We have data on trends in 2-qubit gate speeds. We infer that trends in these gate speeds will be similar across types of quantum gates. Quan-

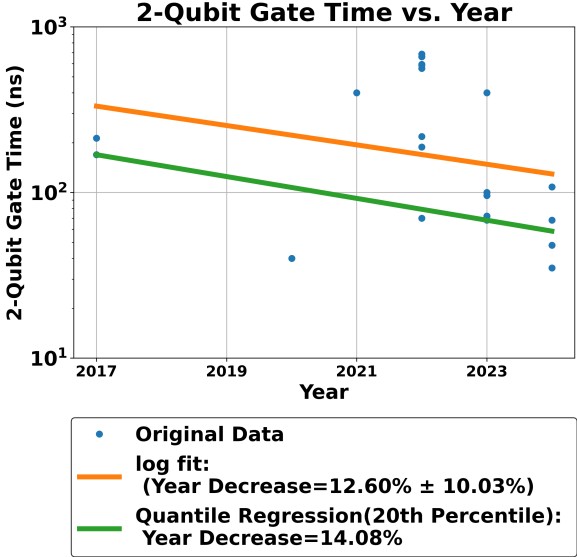

*Figure 6.* Trends in 2-qubit gate times for superconducting quantum computers. Data from Ruane et al. (2025).

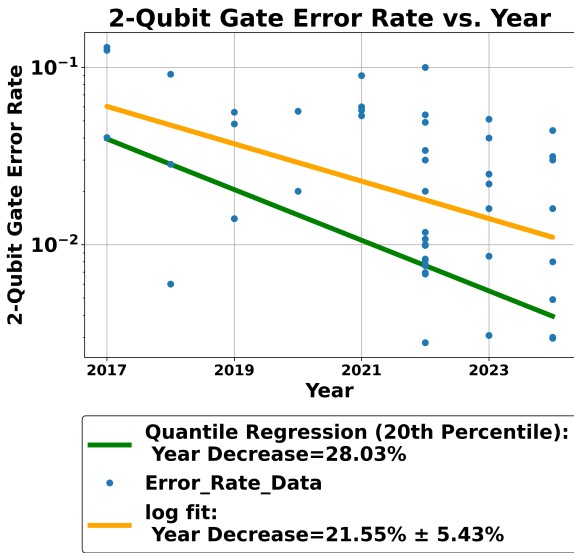

*Figure 7.* Trends in 2-qubit superconducting error rate. Data from Ruane et al. (2025).

tum algorithms are usually measured by the number of Toffoli gates. Three-qubit Toffoli gates generally have much longer gate times than two-qubit gates because they require magic-state distillation, making them roughly 10–100 × slower (Babbush et al., 2021).

### C.3. Quantum Error Correction: Logical vs Physical Qubits

Surface codes are the leading error correction paradigm for superconducting quantum computers (Sevilla & Riedel, 2020). Once the two-qubit gate error rate falls below the error threshold $p_{th} \approx 10^{-2}$, we can use surface codes to correct the gate error rate to arbitrarily low levels. This error correction threshold has only recently been reached for superconducting qubits (Acharya et al., 2024). However, this comes at a high cost. We must use a much higher number of physical qubits to form one error-corrected or logical qubit (a qubit with a significantly low error rate $p_{\mathrm{L}} \approx 10^{-18}$). The number of physical to logical qubits needed for an error rate $p$ is approximately given by Sevilla & Riedel (2020) :

$$f_{\mathrm{QEC}}(p) = \left[ 4 \frac{\log\left(\sqrt{10}p/p_{\mathrm{L}}\right)}{\log\left(p_{\mathrm{th}}/p\right)} + 1 \right]^{-2} \quad (2)$$

We've collected data on trends in 2-qubit gate error rates in Fig 7 from (Ruane et al., 2025). Using equation C.3, along with an exponential extrapolation of trends, we can predict the quantum error correction overhead/physical-to-logical-qubit ratio. We fit a 20th percentile quantile regression to trends in error rate and use the 20th percentile trend as our

default for progress in state-of-the-art chips.

### C.4. How Does Error Correction Affect Quantum Computer Efficiency?

Error correction adds a large overhead to quantum computer efficiency and, therefore, quantum computer speed and price. The physical-to-logical ratio (error correction overhead) is related to the code distance by the following equation (Sevilla & Riedel, 2020) :

$$f_{QEC} = (2d - 1)^2 \quad (3)$$

Given an error correction code distance $d$, a Toffoli gate in a quantum computer requires $5.5d$ surface code cycles. This means the gate time is $\propto \sqrt{f_{QEC}}$ (Babbush et al., 2021). In addition, the total number of Toffoli gates necessary is proportional to the number of physical qubits, which is $f_{QEC}$ times the number of logical qubits (Gidney et al., 2024). Hence, the total number of Toffoli gates necessary in a given quantum algorithm is proportional to $f_{QEC}$. Combining these two factors, we see that the total number of logical operations per second is $\propto f_{QEC}^{-3/2}$. This means that the total quantum overhead combining latency and parallelism factors is also $\propto f_{QEC}^{-3/2}$.

### C.5. Connectivity and Other Hardware Constraints

In our study, we chose not to include a connectivity penalty term. All of the algorithms we address in this paper only need $O(\log(n))$ qubits for a problem of size $n$. For a quantum chip with 2d-connectivity this means there will be an $O(\sqrt{\log(n)})$ overhead (Herbert, 2018). 2d-connectivity is

the most natural form of connectivity for surface code-based error correction (He et al., 2025). However, it is possible to implement the surface code with different connectivity or use different code. This would result in other overheads, but an even smaller asymptotic connectivity overhead.

## D. Maximum Time Constraint

Here, we expand on the robustness analysis outlined in Section 8. We vary the maximum computation allowed from 1 week to 1 year. We see no discernible change within this time period variation. This is shown in Fig 8.

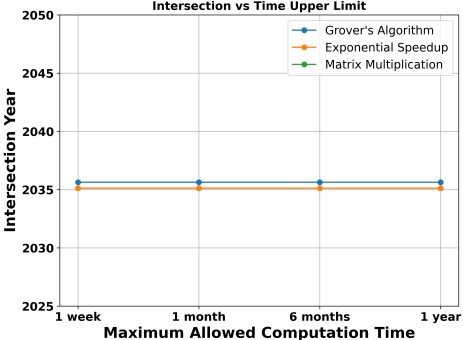

*Figure 8.* Variation of maximum computation time has little effect on the year of first quantum advantage for the algorithms examined in our paper. QEA years past 2050 are not plotted, which is the case for dense matrix multiplication.

## E. Other Types of Quantum Hardware

Most of our data on quantum computers comes from superconducting systems. However, we have also done some evaluation of other hardware like ion-trap and neutral atom quantum computers. Ion-trap quantum computers show growth in qubit numbers similar to superconducting systems (see Fig 10). In addition, ion-trap computers also have a smaller gate error rate, and the gate error is falling substantially faster than superconducting systems (see Fig 9). However, they are more expensive and orders of magnitude slower. For instance, IonQ Aria has 2-qubit physical gate times around half a millisecond (IonQ Staff, 2025), which is about a thousand times slower than superconducting systems and has a price of $10^{-3}$\$ per 2-qubit gate (Microsoft, 2025a), which makes it about a thousand times more expensive. Even with significantly lower error correction overhead, ion-trap systems are at a disadvantage for all the tasks we investigate in this report.

Neutral atoms are an emerging quantum computing platform with significant potential. In particular, they are viewed as a more scalable alternative to superconducting qubits, offering long coherence times (Gschwendtner et al., 2023). Current pricing for neutral atom systems is roughly an order of mag-

nitude lower than that of superconducting platforms—for example, PASQAL charges about 0.10\$ per second of compute time (Microsoft, 2025a). However, few neutral atom providers publicly report two-qubit gate times. According to Wintersperger et al. (2023), a neutral atom system can perform two-qubit CZ gates in approximately 400 ns, which is about an order of magnitude slower than superconducting gates.

Overall, neutral atom quantum computers appear to be a promising and potentially competitive alternative to superconducting systems. Nevertheless, much more experimental data—particularly on gate fidelities and operation speeds—will be needed before rigorous performance comparisons and economic analyses can be made.

In addition to ion-trap and neutral atom systems, there is a wide range of other important quantum hardware. These include spin qubits, quantum annealers, and photonic quantum computers. Unfortunately, we do not have sufficient data to make conclusions about these platforms but do not rule them out as potentially useful for quantum deep learning.

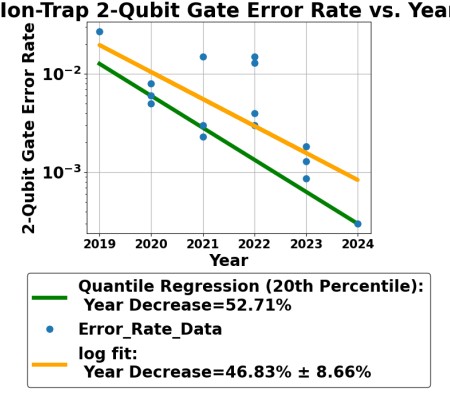

*Figure 9.* Decrease in ion-trap 2-qubit error rate over time. The fall in error rates is approximately twice as fast as superconducting qubit systems. Data sourced from Ruane et al. (2025).

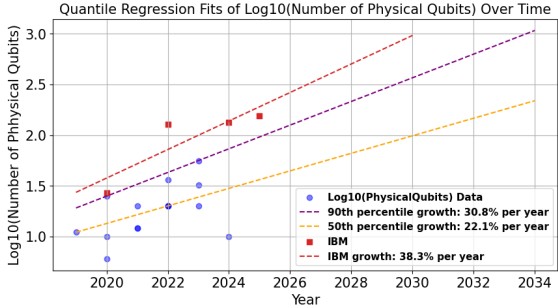

*Figure 10.* Increases in the number of physical ion-trap qubits over time in comparison to the growth rate of IBM-superconducting qubits in red. Overall growth rates for superconducting and ion-trap systems are similar. Data sourced from Ruane et al. (2025).

# F. HHL Criteria For Exponential Advantage

Given the importance of the HHL algorithm, we would like to elaborate on the algorithm and the conditions necessary for exponential quantum advantage. HHL solves the quantum linear algebra problem, which is different from the traditional linear algebra problem. The quantum linear algebra problem is focused around:

$$A \ket{x} = \ket{b}$$

where we are interested in finding $\ket{x}$. $\ket{x}$ is a normalized quantum state corresponding to the full solution $x$. $A$ is assumed to be an $n \times n$ hermitian matrix, with condition number $k$, and at most $s$ nonzero entries in any row or column. Retrieving an approximate $\ket{\widetilde{x}}$ to $\ket{x}$ can be done in time $O(\log(n)\kappa s \text{polylog}(\kappa s/\epsilon))$ with accuracy $\|\ket{\widetilde{x}} - \ket{x}\| \leq \epsilon$ under the conditions below (Zlokapa et al., 2021). This state can be used to measure $\bra{x}M\ket{x}$ for some linear operator M. The resulting output can be done classically in time $O(ns\kappa \log(1/\epsilon))$ under similar conditions using conjugate gradient methods (Dervovic et al., 2018). If we would like to maintain an exponential advantage over a classical computer. We must meet the following criteria:

**Aaronson's Criteria (Aaronson, 2015):**

1. The vector $\ket{b}$ must be efficiently preparable. This usually assumes QRAM, but QRAM is not a necessary or sufficient condition.

2. The matrix $A$ must be $s$-row sparse. This means that it has at most $s$ nonzero elements per row and s must be at most $O(\text{polylog}(n))$.

3. The matrix $A$ must have condition number $\kappa$. To maintain an exponential advantage, $\kappa$ must be polylogarithmic in $n$.

4. HHL only returns $\ket{x}$. Returning the full vector $x$ leads to an overhead of order $O(n)$.

**Rank Criteria:**

1. For exponential advantage, $A$ must not be low-rank. This is due to dequantized algorithms which are able to implement low-rank matrix inversion in polylogarithmic time in the dimension $n$ (Chia et al., 2018). This means the rank of A must be at least $\Omega(poly(n))$ (Zlokapa et al., 2021). These polynomial exponents are usually quite high, so we can neglect this condition in most analyses of quantum advantage.

# G. Code and Data

Our code is available here: https://github.com/hansgundlach/QuantumDL. This is sufficient to reproduce the figures in the main body. Our analysis includes superconducting qubit trend data, but we are not able to publicize the dataset at this point in time.

