# OpenReview forum: "Position: Quantum Deep Learning Still Needs a Quantum Leap"
_ICML.cc/2026/Position_Paper_Track — ICML 2026 Position Paper Track regular_

### Official Review · Reviewer_K22L · 2026-02-15

**Significance:** 3
**Argument Clarity:** 2
**Rating:** 4
**Confidence:** 3

**Questions:**

N/A

**Alternative Views Section:**

Yes

**Compliance With Llm Reviewing Policy A Conservative:**

Affirmed.

**Discussion Potential:**

2

**Final Justification:**

After reading the rebuttal by the authors, I appreciate the authors’ clarification and their willingness to improve the paper’s organization and readability. While some sections still move quickly over complex literature, I do not see major issues overall and now lean toward acceptance.

**Paper Summary:**

This paper argues that even accounting for the advancements in quantum computing technology, a quantum leap would be needed for quantum computers to meaningfully impact deep learning over the coming decade or two.

**Position:**

Yes

**Position In Title:**

Yes

**Related Work:**

2

**Strengths And Weaknesses:**

Strengths

1. The gap between quantum computing and deep learning is an important issue to study for the QML community.

2. Three important areas are discussed to support the position of the authors.

Weaknesses

(Major) 1.  Although the paper 'reveals three important areas where quantum computing could potentially accelerate
deep learning, each of which faces a challenging roadblock to realizing its potential', the arguments for each 'area' appear scattered throughout the article rather than forming a structured line of reasoning. This makes the paper hard to read and the arguments hard to follow.  The paper should be reorganized into a clearer structure to make it easier to follow.

(Minor) 2.  The paper assumes algorithmic constants = 1 for both quantum and classical algorithms, which might be unfair for some approaches.

(Minor) 3. Some typos in the paper, e.g., Sec 3.1.

**Support:**

2

---

> ### Author Rebuttal · Authors · 2026-03-31
>
> **(Major) 1. Paper organization and clarity of argument:**
> We considered organizing the paper into three sections corresponding to the three roadblocks we identified. However, we felt this would overemphasize the negative conclusions at the expense of presenting each algorithm's unique potential alongside its limitations, which we believe is important for a balanced position paper. That said, we agree the current structure could benefit from a clearer throughline. To address this, we plan to add a summary table to the camera-ready version listing each algorithm, its potential speedup regime, the roadblock category it falls under, the corresponding QEA threshold, and our overall assessment. We believe this will substantially improve readability and allow the reader to track the argument across sections without losing sight of the overarching structure.
>
> **(Minor) 2. Assumption of algorithmic constants = 1:**
> We address this concern in detail in our response to Reviewer Eni3 (Weakness 1). To briefly summarize here: our robustness analysis (Figure 7, currently in Appendix D; we plan to move it to the main body) shows that our conclusions are robust to constant-factor changes spanning 10⁻³ to 10³. When these perturbations do shift our findings, they generally make the outlook more pessimistic for quantum algorithms, maintaining our overall conclusion.
>
>
> **(Minor) 3. Typos:**
> Thank you for catching this. We will correct these in the revised manuscript.

---

> > ### Author Rebuttal · Reviewer_K22L · 2026-04-01
> >
> > I agree with 7zQq that the survey is broad, but some sections move too quickly over complex literature. The paper sacrifices readability by trying to cover too much. Still, I do not see major issues with the paper overall. Since the authors have acknowledged these concerns and committed to making revisions, I now lean positive.

---

### Official Review · Reviewer_cSM9 · 2026-03-04

**Significance:** 4
**Argument Clarity:** 4
**Rating:** 5
**Confidence:** 2

**Questions:**

As a reader with limited knowledge of quantum computing, I'd like to humbly ask

1. Given that current quantum computers are significantly slower than traditional GPUs in single-step operations, is it possible that a completely new algorithm will emerge in the future, one that doesn't need to compete with traditional computers in terms of speed, but instead uses a completely different approach to solve machine learning problems?

2. The article mentions the need to accelerate the development of QRAM. In your opinion, as deep learning practitioners, is it necessary to learn about quantum computing now, or should we wait until the hardware is more mature?

**Alternative Views Section:**

Yes

**Compliance With Llm Reviewing Policy A Conservative:**

Affirmed.

**Discussion Potential:**

4

**Final Justification:**

The authors have addressed my concerns, I confirm my evaluation and maintain the score.

**Paper Summary:**

This is a position paper that primarily discusses whether quantum computing can truly be used to accelerate deep learning within the next ten to twenty years. The authors reviewed several quantum algorithms and found that while they theoretically reduce the number of computation steps, quantum computers are actually too slow in practice. Furthermore, many promising quantum algorithms require a data retrieval technique called QRAM, which is currently very immature. Additionally, some algorithms only perform well under very demanding and specific conditions. Therefore, the authors believe that quantum deep learning still needs a massive leap forward to become truly useful.

**Position:**

Yes

**Position In Title:**

Yes

**Related Work:**

3

**Strengths And Weaknesses:**

Strengths:
To be honest, I'm not an expert in quantum computing, and I don't fully understand the complex derivations and quantum physics involved. However, I found this article relatively easy to understand. The author didn't exaggerate the power of the new technology; instead, they realistically listed the specific problems we currently face, such as slow hardware speeds and difficulties in data loading. This made it very clear to me that there are still so many real obstacles to applying quantum computing to ordinary deep learning.

Weaknesses:
Because of my limited professional background, I find it difficult to verify whether the author's predictions about hardware development (such as the mentioned $10^{13}$ deceleration rate) are entirely accurate. Furthermore, the article as a whole reads somewhat pessimistically. Perhaps the author could have elaborated a bit more at the end: what can ordinary machine learning researchers do to prepare for the long wait for a breakthrough in quantum hardware? Or can we gain some inspiration from the ideas in quantum algorithms? This might be more helpful to everyone.

**Support:**

3

---

> ### Author Rebuttal · Authors · 2026-03-31
>
> **Weakness (Broader guidance):**
> There are several broader points that are valuable to general deep learning practitioners, and we outline some in our responses to Question 2 below. In short, deep learning practitioners can concentrate on current GPU-based workflows without worrying about quantum algorithms replacing them in the near future. That said, quantum-inspired classical algorithms have already produced practical gains (see our response to Q2), and we will expand the call to action in our revision to highlight these cross-pollination opportunities more clearly.
>
> **Question 1 (Could a fundamentally different quantum approach bypass the speed limitation?):**
> Some current quantum algorithms already take quite different approaches to learning, and further developments are certainly possible. However, any new approach will still need to compete on correctness and performance. Quantum computers could also enable data analysis on quantum datasets in ways that are not classically possible. That said, we don't yet see papers laying out clear applications of these abilities to practical problems like material design. We think this is an important open direction and will flag it more prominently in the revision.
>
> **Question 2 (Should ML practitioners learn quantum computing now?):**
> Not at present. In the same way that current ML practitioners do not need to learn other emerging computing paradigms (analogue, stochastic, reversible computing), quantum ML will likely remain a specialized subfield for some time. However, we'd add one important caveat: there are cases like [Tang (2018)](https://arxiv.org/abs/1807.04271) where quantum algorithms served as direct inspiration for new classical data analysis methods. Tensor networks are another example: originally developed for quantum many-body physics, they are now actively used for neural network compression and efficient parameterization of large models. So while deep familiarity with quantum hardware is unnecessary, keeping an eye on the algorithmic ideas coming out of the field can be practically useful.

---

> > ### Author Rebuttal · Reviewer_cSM9 · 2026-04-01
> >
> > Got it. I will maintain my evaluation and score.

---

### Official Review · Reviewer_Eni3 · 2026-03-11

**Significance:** 4
**Argument Clarity:** 3
**Rating:** 4
**Confidence:** 4

**Questions:**

1.	The QEA model assumes specific hardware-overhead trends and time constraints. How sensitive are the conclusions to moderate deviations in slowdown factors (e.g., one to two orders of magnitude improvement beyond projections)? Would such shifts materially change the 10-20 year outlook?
2.	In Section 5 (matrix multiplication and attention), the argument focuses on dense-output requirements. Could hybrid quantum-classical models, where only compressed or partial statistics are extracted, alter the feasibility boundary in meaningful ways?

**Alternative Views Section:**

Yes

**Compliance With Llm Reviewing Policy A Conservative:**

Affirmed.

**Discussion Potential:**

4

**Final Justification:**

In light of the authors' rebuttal and other reviews, my score is confirmed and I lean towards acceptance.

**Paper Summary:**

This manuscript focuses on the broader context in which quantum computing is frequently proposed as a transformative accelerator for deep learning. The paper advances the position that, despite rapid hardware progress and theoretical algorithmic speedups, meaningful practical impact on deep learning is unlikely within the next one to two decades without substantial breakthroughs. The argument is supported by a quantitative “Quantum Economic Advantage” (QEA) model extending prior work, applied across clustering, linear algebra, Grover-based search, and neural network training scenarios. Overall, this paper addresses an important concept by systematically identifying bottlenecks (most notably hardware overhead, QRAM limitations, and restrictive algorithmic assumptions) and outlining research directions necessary to overcome them.

**Position:**

Yes

**Position In Title:**

Yes

**Related Work:**

4

**Strengths And Weaknesses:**

Strengths

1.	The paper explicitly articulates a position (that quantum deep learning requires major breakthroughs before delivering practical advantage) and maintains consistency throughout. The stance is evident in the title and is reinforced throughout the sections.
2.	The work synthesizes a broad range of quantum algorithms (e.g., HHL-based methods, Grover search, quantum clustering, matrix multiplication, attention mechanisms) and integrates them into a unified forecasting framework. The QEA model, illustrated in Figures 1-3, compares asymptotic advantage against hardware constraints.
3.	The paper includes a substantive “Alternative Views” section (Section 8), addressing hardware breakthroughs, classical slowdowns, NISQ-era possibilities, and algorithmic shifts.
4.	The topic intersects current enthusiasm around quantum ML. By presenting a skeptical but data-driven position, the paper is likely to stimulate meaningful discussion.

Weaknesses

1.	The QEA framework depends on several strong simplifications (e.g., constant factors set to 1, extrapolated qubit trends, long time constraints). Although robustness is briefly claimed, the sensitivity of conclusions to these assumptions is not explored in sufficient detail.
2.	The argument relies heavily on theoretical scaling and projected hardware trends. There is no experimental or simulation-based validation of the QEA thresholds on realistic workloads. Even small empirical case studies (e.g., simulated cost comparisons for matrix multiplication at defined scales) would significantly strengthen the position.

**Support:**

3

---

> ### Author Rebuttal · Authors · 2026-03-31
>
> Thank you for the detailed review. We address each point below.
>
> **Weakness 1 (Sensitivity of conclusions to simplifying assumptions):**
> We appreciate this feedback; we should have emphasized our robustness analysis more prominently. Figure 7 (currently in Appendix D) presents a detailed sensitivity analysis across our model parameters; we plan to move it into the main body for the camera-ready version. Our conclusions are robust to constant-factor changes spanning 10⁻³ to 10³. Notably, when parameter perturbations do shift our findings, they generally make the outlook more pessimistic for quantum algorithms, not less. The one notable exception is gate-time improvement rate: substantially faster progress (~40% per year vs. the ~13% we currently measure) could bring quantum matrix multiplication benefits into the 2040s.
>
> **Weakness 2 (Lack of experimental or simulation-based validation):**
> Our paper focuses on fault-tolerant quantum algorithms, where we can make rigorous asymptotic claims across broad problem classes. No medium- or large-scale fault-tolerant quantum computers currently exist, making direct empirical validation infeasible at this time. For near-term (NISQ) quantum ML algorithms, extensive experimental benchmarking has been conducted by others. We summarize some of these results in our paper, and the findings are largely negative or mixed (see e.g., [Bowles et al., 2024](https://arxiv.org/abs/2403.07059); [Bermejo et al., 2024](https://arxiv.org/abs/2408.10274)). To our knowledge, these near-term algorithms are not currently deployed in industry. We will add a clearer signpost to this existing literature in the revised manuscript.
>
> **Question 1 (Sensitivity to moderate deviations in slowdown factors):**
> We refer the reviewer to our response to Weakness 1 and Appendix D above, where we address this directly. Our model outcomes are robust to constant-factor deviations of several orders of magnitude in the slowdown factor. Only under aggressive gate-time improvement assumptions (greater than 40%/year) do we see the outlook shift materially, bringing quantum matrix multiplication advantages into the 2040s. Therefore, a one-to-two orders of magnitude overhead improvement beyond projections would not change the 10–20 year outlook under our model.
>
> **Question 2 (Hybrid quantum-classical models with partial output extraction):**
> In cases where only partial statistics are required, quantum algorithms do look more feasible. However, it is difficult to identify algorithms that are both clearly highly capable for deep learning workloads and avoid dense intermediate conversions or dense outputs in matrix-related computations. Quantum ansatz-based algorithms avoid this requirement but suffer from the limitations discussed in our response to Weakness 2 (i.e., mixed/negative experimental results for near-term approaches). Other models require nonlinearities that cannot be performed natively on a quantum computer, reintroducing dense output requirements in the processing pipeline. We do analyze algorithms like Grover's that require only limited output. We would welcome suggestions of specific algorithms that represent important advances for deep learning while requiring only limited output. This would be a valuable direction for future work.

---

> > ### Author Rebuttal · Reviewer_Eni3 · 2026-04-04
> >
> > I would like to thank the authors for the response. I will consider adjusting my score accordingly.

---

### Official Review · Reviewer_7zQq · 2026-03-13

**Significance:** 2
**Argument Clarity:** 3
**Rating:** 4
**Confidence:** 4

**Questions:**

1. The central forecasts rely on hardware-overhead assumptions. How sensitive are the paper’s main qualitative conclusions to more aggressive assumptions about photonic or other non-superconducting quantum architectures?
2. The paper repeatedly treats QRAM as a core bottleneck. Is the authors’ claim that practical QRAM is unlikely in the relevant time horizon, or that even with QRAM many major DL use cases would still remain unattractive?
3. The paper is skeptical of claimed exponential training advantages due to restrictive assumptions. Which assumptions do the authors view as the real deal-breakers: QRAM, condition numbers, sparsity, data access, output-readout, or something else?

**Alternative Views Section:**

Yes

**Compliance With Llm Reviewing Policy A Conservative:**

Affirmed.

**Discussion Potential:**

3

**Final Justification:**

The authors have addressed my concerns

**Paper Summary:**

This position paper argues that quantum computing is unlikely to deliver practical benefits for mainstream deep learning in the next 10–20 years without major further breakthroughs. The authors’ main point is that many claimed quantum advantages look appealing asymptotically, but become much less convincing once one accounts for realistic hardware overhead, limited logical qubits, data-loading costs, and end-to-end runtime constraints.
To support this position, the paper reviews several candidate application areas—including clustering, hyperparameter search, reinforcement learning, matrix multiplication, attention, classical training, and quantum neural networks—and argues that most either offer only modest speedups, rely on strong assumptions such as QRAM, or apply only in narrow settings. The overall conclusion is that quantum deep learning remains scientifically interesting, but still lacks the algorithmic and hardware breakthroughs needed for broad practical impact.

**Position:**

Yes

**Position In Title:**

Yes

**Related Work:**

3

**Strengths And Weaknesses:**

Strengths
1. The paper addresses a question that is currently receiving substantial hype: whether quantum computing will materially accelerate AI/deep learning. That makes the position highly relevant.
2. The claim is easy to understand and consistently maintained throughout the paper: many quantum speedups do not survive end-to-end practical constraints.
3. The quantum economic advantage framing is a strong contribution for a position paper because it moves the discussion away from asymptotic slogans and toward feasibility under hardware and time constraints.
4. The paper does not focus on a single niche method; it covers preprocessing, clustering, hyperparameter search, RL, matrix multiplication, attention, classical training, and quantum neural networks. That breadth is valuable in a position track paper.

Weaknesses
1. The conclusions depend heavily on the forecasting model. Although the model is sensible, many conclusions hinge on assumptions about hardware trends, constants, fault tolerance, and what counts as realistic ML runtimes. A position paper can do this, but the claims inherit that uncertainty.
2. In several places, the paper compares asymptotic quantum proposals to current classical practice, but the “practical classical baseline” is not always equally developed or optimized. That can make the practical disadvantage of quantum methods feel stronger than rigorously established.
3. The survey is wide, but some sections move quickly over complicated literatures. For example, the treatment of VQAs and quantum neural networks is necessarily high-level, which weakens the force of any strong negative conclusion there.
4. The paper is strongest at critique, but weaker at positive guidance. The call to action is sensible, but somewhat generic: better QRAM, better characterization, better algorithms. It would be stronger if it more sharply prioritized research directions most likely to change the conclusion.

**Support:**

2

---

> ### Author Rebuttal · Authors · 2026-03-31
>
> Thank you for your detailed feedback. We address each point below.
>
> **W1: Dependence on forecasting model.**
> Many aspects of our analysis are hard to pin down exactly. However, our conclusions are generally robust to many variations we tested.  To make this more transparent, we plan to move the robustness analysis (currently in Appendix D) into the main paper. This analysis shows that our qualitative conclusions are stable across a wide range of parameter variations and runtime constraints (e.g., varying constants from 10⁻³ to 10⁴, and varying hardware trend growth rates). We also note that we use asymptotic times for current algorithms, which may improve in the future, an important caveat we will make more prominent.
>
> **W2: Classical baselines not equally optimized.**
> We deliberately compare against the classical algorithms actually used (or likely to be used) in deep learning practice, rather than the theoretically best classical algorithms. For example, for matrix multiplication we compare to the standard $O(N^{3})$ algorithm rather than Strassen's $O(N^{2.8})$, since Strassen is rarely used in deep learning frameworks. Similarly, we reference the standard quadratic attention mechanism rather than linear variants, since these do not yet fully replace their quadratic counterparts in production systems. We will clarify these choices in revisions.
>
> **W3: High-level treatment of VQAs.**
> We acknowledge that our discussion of VQAs is necessarily condensed given page constraints and the scope of our quantum advantage model. We believe there has been excellent discussion elsewhere in the literature. In particular, we point readers to [Bowles et al. (2024), "Better than Classical? The Subtle Art of Benchmarking Quantum Machine Learning Models"](https://arxiv.org/abs/2403.07059). We will add a more explicit pointer to and a summary of this work to help readers engage with the VQA literature in greater depth.
>
> **W4: Strengthening the call to action.**
> In revision, we will sharpen our prioritization of research directions. Specifically, we plan to:
> - Emphasize the development of QRAM-aware quantum algorithms. We are particularly impressed by work like [Harrow (2020), "Small Quantum Computers and Large Classical Data Sets"](https://arxiv.org/abs/2004.00026), which achieves speedups without requiring full QRAM.
> - Highlight approaches like [Liu et al. (2024)](https://arxiv.org/abs/2303.03428) that include QRAM and non-QRAM approaches and engage with contemporary deep learning architectures, while noting that conditions like "sufficiently dissipative" need much better characterization.
> - Add a summary table mapping each use case to its specific bottlenecks, making the prioritization more concrete.
>
> **Q1: Sensitivity to non-superconducting architectures (e.g., photonic).**
> Our robustness analysis (Appendix D) varies key parameters across wide ranges (1000x larger and smaller), and we compare to ion-trap and neutral atom systems in Appendix E. For photonic quantum computing, we currently lack sufficient empirical data on gate speeds, error rates, and scalability to make confident quantitative claims. We acknowledge this as a limitation and flag photonic systems as a potentially important avenue that could alter our conclusions, particularly for problems where gate speed is the primary bottleneck (e.g., matrix multiplication). We will clarify this in revision.
>
> **Q2: QRAM as a bottleneck.**
> Our position is that QRAM is one of several bottlenecks, not the sole one. Even assuming the availability of large-scale, fault-tolerant QRAM, many quantum algorithms would still be too slow to justify commercial use due to the hardware overhead. We will clarify this distinction: QRAM is necessary but not sufficient for practical quantum advantage in many deep learning applications.
>
> **Q3: Which assumptions are the real deal-breakers for exponential training advantages?**
> The most restrictive deal-breaker is probably the absence of QRAM. Many deep learning workflows involve analyzing classical data, and this is limited or impossible without QRAM. We think algorithm developers, specifically those at conferences like ICML, should develop more QRAM-aware algorithms, and there should be greater recognition of this issue among the broader community interested in the potential of new hardware paradigms for deep learning. Large overhead is also a significant restriction, but there could be faster paradigms like photonic quantum computing. Further, current quantum computers are heavily optimized for fault tolerance; if this were no longer a binding constraint, we could see larger speed improvements. Finally, as discussed above, the restrictive conditions in HHL-based algorithms (sparsity, condition number, readout) and in specific proposals like "sufficient dissipation" ([Liu et al., 2024](https://arxiv.org/abs/2303.03428)) or "sufficient clusterability" (q-means) remain hard to characterize for real deep learning workloads.

---

> > ### Author Rebuttal · Reviewer_7zQq · 2026-04-07
> >
> > I would like to thank the authors for the response.
> > I will adjust my score accordingly.

---

### Decision · Program_Chairs · 2026-04-30

**Decision:**

Accept (regular)

**Comment:**

The paper received four positive leaning reviews (1 accept, three borderline accepts). There was general appreciation for the importance (and timeliness) of the paper's position, the breadth of scope, and the substantive alternative views. There were some concerns raised during the review process, majority of which were addressed by the authors' response. Therefore, an accept consensus was reached.